# Clinicoprognostic and Histopathological Features of Guttate and Plaque Psoriasis Based on PD-1 Expression

**DOI:** 10.3390/jcm10215200

**Published:** 2021-11-07

**Authors:** Chang-Jin Jung, Hee-Joo Yang, Seung-Hyun Bang, Woo-Jin Lee, Chong-Hyun Won, Mi-Woo Lee, Youngsup Song, Sung-Eun Chang

**Affiliations:** 1Department of Dermatology, Asan Medical Center, University of Ulsan College of Medicine, 88 Olympic-ro 43-gil, Songpa-gu, Seoul 05505, Korea; juchji630@naver.com (C.-J.J.); heejoo1210@naver.com (H.-J.Y.); bangsh1037@gmail.com (S.-H.B.); uucm79@hanmail.net (W.-J.L.); chwon98@chol.com (C.-H.W.); miumiu@amc.seoul.kr (M.-W.L.); 2Department of Biomedical Sciences, Asan Medical Center, University of Ulsan College of Medicine, 88 Olympic-ro 43-gil, Songpa-gu, Seoul 05505, Korea; ysong@amc.seoul.kr

**Keywords:** psoriasis, epidermal PD-1, dermal PD-1, immune checkpoint-related markers, clinicoprognostic characteristics

## Abstract

Several studies have determined the correlation between programmed cell death protein-1 (PD-1) and chronic plaque psoriasis (CPP). However, limited studies have assessed the association between PD-1 expression and the clinicoprognostic and distinct clinicopathological characteristics of CPP and guttate psoriasis (GP). Twenty-nine patients with skin biopsy-confirmed CPP were recruited at the Asan Medical Center between January 2018 and June 2020, and 33 patients with biopsy-confirmed GP were enrolled between January 2002 and June 2020. The clinicoprognostic and histopathological characteristics were analyzed according to immunohistochemical PD-1 expression in the epidermal or dermal inflammatory infiltrates. The CPP and GP lesions were divided into PD-1-low and PD-1-high groups. The CPP epidermal PD-1-high group had typical histopathological changes and significantly higher psoriasis area and severity index scores (*p* = 0.014) and disease duration (*p* = 0.009) than the epidermal PD-1-low group. In patients with GP, compared with the dermal PD-1-high group, the dermal PD-1-low group exhibited significantly higher disease duration (*p* = 0.002) and relapse rate of plaque psoriasis (*p* = 0.005) and significantly lower relapse-free survival (*p* = 0.016). Upregulated epidermal PD-1 expression was correlated with the chronicity and severity of CPP, while downregulated dermal PD-1 expression was correlated with poor prognosis of GP.

## 1. Introduction

Psoriasis is a chronic, recurrent inflammatory disease associated with the activation of T cells, including T helper 17 (Th17) and Th1 cells, which promote cutaneous inflammation and keratinocyte hyperproliferation [1]. Interleukin (IL)23/IL17 is the major effector cytokine axis involved in the pathogenesis of psoriasis. Additionally, the pathogenesis of psoriasis is mediated by pro-inflammatory cytokines, including IL6, TNF-α, and IL1.

Dysfunctional regulatory T (Treg) cells disrupt immune tolerance, which leads to the induction of autoimmune and auto-inflammatory diseases, such as psoriasis [2]. In addition to the immune-suppressing cytokines released from Treg cells, some T cell surface molecules, including cytotoxic T lymphocyte-associated antigen 4 (CTLA4), neurophilin-1, human leukocyte antigen G, and the programmed cell death protein-1 (PD-1) that interacts with the programmed cell death ligand (PD-L1), play critical roles in immune tolerance [3]. Dysfunctional T cell surface molecules cannot inhibit the activity of inflammatory cells in patients with psoriasis. In particular, the role of PD-1 in psoriasis has piqued the interest of the scientific community. PD-1, a glycoprotein expressed on the surface of various immune cells, including T cells, B cells, macrophages, and monocytes, binds to PD-L1 and PD-1LG2 and consequently inhibits the function of effector T cells and promotes Treg cell activity [4]. Previous studies have reported that the PD-1 signaling pathway modulates the production of cytokines, such as interferon γ (IFN-γ), IL2, IL17, and tumor necrosis factor α (TNF-α). Consequently, PD-1 regulates disease pathogenesis by modulating the Th1 and Th17 axes [4,5]. In murine models, PD-1 deficiency induces psoriasiform dermatitis, promotes the recruitment of activated cytotoxic CD8+ T cells in the epidermis, and consequently upregulates the production of IL6 [6]. Recombinant PD-1 treatment alleviated psoriatic inflammation in a murine model of imiquimod-induced psoriasis [7]. 

Guttate psoriasis (GP) is characterized by a sudden onset of widely dispersed scaly erythematous papules that are less than 1.5 cm in diameter [8]. The distinct clinical characteristics of GP include the involvement of human leukocyte antigen (HLA)-Cw6 and a high prevalence of preceding streptococcal infection [9]. Compared with chronic plaque psoriasis (CPP), GP is likely associated with rapid involution and good prognosis. However, patients may exhibit GP relapse or plaque-type psoriasis even after spontaneous involution [10]. Only a few studies have focused on the differential immunological pathogenesis of CPP and GP.

This study aimed to comparatively examine the clinicoprognostic and histopathological characteristics of patients with CPP and GP according to the immunohistochemical (IHC) expression levels of PD-1.

## 2. Materials and Methods

### 2.1. Patient Selection

This retrospective study was approved by the Institutional Review Board (IRB) of the Asan Medical Center (IRB No. 2020-0862). 

Twenty-nine patients who were diagnosed with CPP through skin lesion biopsy at the Asan Medical Center between January 2018 and June 2020 were included in this study. All patients provided written informed consent for publication of their case details. Among the 29 CPP patients, non-lesional skin biopsy samples could be obtained in part of them, only 11 cases, which is one of limitations of our study. During skin biopsy, lesional and non-lesional discarded superficial skin fragments (approximately 0.5 mm in size) were stored in RNA lysis buffer for mRNA expression levels of PDL-1. 

Thirty-three patients who were diagnosed with GP through a clinicopathological analysis of their skin lesion biopsies were recruited at the Asan Medical Center between January 2002 and June 2020. The exclusion criteria for patients with GP were as follows: acute flare-up of other types of preexisting psoriasis and loss of contact during the collection of data on clinical course, including psoriasis relapse and extent of skin lesions. 

### 2.2. Demographic and Clinical Characteristics

Demographic and clinical data, including age at diagnosis, sex, family history of psoriasis, history of previous upper respiratory infection (URI), disease grades (psoriasis area and severity index (PASI) and body surface area (BSA)), presence of pruritus, treatment modalities, and disease duration, were collected.

Additional prognostic clinical data, including time to disease resolution after treatment, GP relapse, and plaque psoriasis relapse, were collected from patients with GP. Relapse-free survival (RFS) was defined as the duration from the date of the resolution of GP to the date of overall psoriasis relapse (both GP and plaque psoriasis). RFS-G and RFS-P were defined as the duration from the date of the resolution of GP to the date of GP and plaque psoriasis relapse, respectively.

The following histopathological characteristics were examined: epidermal thickness, horny layer thickness, rete ridge count, grade of inflammatory cellular infiltration, and grade of papillary dermal vessel dilatation. Kim et al. [11]. reported that the severity of dilatation was correlated with the clinical severity of psoriasis. The grades of cellular infiltration and vessel dilatation were scored as follows: 0, absent; 1, mild; 2, moderate; and 3, severe.

### 2.3. IHC Analysis

The tissues obtained for routine diagnostic pathological examinations were used for IHC analysis. The IHC analysis was performed using anti-PD-1 antibodies (mouse monoclonal; clone MRQ-22 [1:1000], Cat# 315M-96, Cell Marque, Rocklin, CA, USA), anti-PD-L1 IHC 22C3 pharmDx (Code SK006, Agilent, Santa Clara, CA, USA), anti-CD4_LSAB (mouse monoclonal; clone SP35 [1:16], Cat# 790-4423, Ventana, Tusan, USA) and anti-CD8 antibody (mouse monoclonal; clone C8/144B [1:100], Cat# 108M-96, Cell Marque, CA, USA).

The formalin-fixed and paraffin-embedded tissue sections were subjected to IHC analysis, with the anti-PD-1 antibody using the ultraView universal alkaline phosphatase red detection kit and BenchMark XT automatic immunostaining device (Ventana Medical Systems), following the manufacturer’s instructions. PD-1-positive staining in more than 50% of the inflammatory cell infiltrate was defined as PD-1-high expression. Patients were thus divided into the following two groups depending on the epidermal expression levels of PD-1: epidermal PD-1-low and epidermal PD-1-high. Similarly, patients were divided into the following two groups according to the dermal expression levels of PD-1: dermal PD-1-low and dermal PD-1-high. The clinicoprognostic and histopathological characteristics were analyzed according to the IHC expression levels of PD-1 in the epidermal or dermal inflammatory infiltrates of patients with CPP. The GP lesions were similarly analyzed for comparison. 

### 2.4. Quantitative Real-Time Polymerase Chain Reaction (qRT-PCR)

The mRNA expression levels of *S100A8* (signature gene of psoriasis) and *PD-L1* in the lesional and non-lesional skin superficial tissues of 11 patients with CPP were analyzed using qRT-PCR. Total RNA was extracted using the total RNA Mini kit (Favogen, Pingtung, Taiwan). The RNA (1 μg) was reverse transcribed into complementary DNA using a RevertAid first-strand cDNA synthesis kit (Thermo Scientific). The qRT-PCR analysis was performed using LightCycler^®^ 480II with AMPIGENE qPCR Green Mix (Enzo Life Sciences, Farmingdale, NY). The PCR conditions were as follows: 95 °C for 5 min (initial denaturation), followed by 45 cycles of 95 °C for 10 s, 60 °C for 10 s, and 72 °C for 10 s. The primer sets used in the qRT-PCR analysis are listed in the Appendix A.

### 2.5. Statistical Analysis

The categorical variables of clinicopathological characteristics according to PD-1 expression levels were assessed using the chi-squared test and Fisher’s exact test. The continuous variables of the clinicopathological characteristics according to PD-1 expression levels were assessed using the Mann–Whitney U test. The correlation between the continuous variables of the clinicopathological characteristics and mRNA expression levels was analyzed using Pearson’s correlation test. RFS was analyzed using the Kaplan–Meier method and the log-rank test. All statistical analyses were performed using GraphPad Prism (version 8.0, GraphPad Software). The differences were considered significant at *p* < 0.05.

## 3. Results

### 3.1. Clinical and Histopathological Characteristics of CPP according to the Levels of PD-1 Expression

The mean age of 29 patients (21 men and 8 women) with CPP was 46.00 years (range, 12–81 years). Of these 29 patients, 14 (48.3%) and 15 (51.7%) patients were assigned to the epidermal PD-1-low (Figure 1A) and epidermal PD-1-high groups (Figure 1B), respectively. However, among the 33 patients with GP, epidermal PD-1 expression was only detected in 1 patient (3.0%) (Figure 1C). The T cell landscape in the representative patients with CPP in the epidermal PD-1-high group is illustrated in Appendix A. While the majority of T cells in the dermis were positive for CD4, the majority of T cells in the epidermis were positive for CD8, expressing PD-1 together.

The demographic data and clinical characteristics of patients with CPP according to epidermal PD-1 expression levels are summarized in Table 1. Sex distribution (*p* = 0.298), age (*p* = 0.949), family history of psoriasis (*p* = 0.483), and preceding URI (*p* = 0.483) were not significantly different between the epidermal PD-1-high and epidermal PD-1-low groups. The PASI score in the epidermal PD-1-high group (mean ± standard deviation (SD), 15.71 ± 9.77) was significantly higher than that in the epidermal PD-1-low group (mean ± SD, 8.20 ± 4.83) (*p* = 0.014) (Figure 1D). The prevalence of pruritus was not significantly different between the two groups (*p* = 0.125). Disease duration in the epidermal PD-1-high group (mean ± SD, 215.2 ± 128.2 months) was significantly higher than that in the epidermal PD-1-low group (mean ± SD, 96.79 ± 122.0 months) (*p* = 0.009) (Figure 1E). The histopathological characteristics of patients with CPP according to the levels of epidermal PD-1 expression are summarized in Table 2. The thickness of the epidermis in the epidermal PD-1-high group (mean ± SD, 289.88 ± 88.88 µm) was significantly greater than that in the epidermal PD-1-low group (mean ± SD, 200.19 ± 20.74 µm) (*p* = 0.004) (Figure 1F). The horny layer thickness (*p* = 0.201), rete ridge count (*p* = 0.354), and grade of inflammatory cellular infiltration (*p* = 0.567) were not significantly different between the two groups. The grade of vessel dilatation in the epidermal PD-1- high group (mean ± SD, 2.07 ± 0.26) was significantly higher than that in the epidermal PD-1-low group (mean ± SD, 1.50 ± 0.52) (*p* = 0.002) (Figure 1G).

The demographic data, clinical characteristics, and histopathological findings of patients with CPP according to the dermal PD-1 expression levels are summarized in Appendix A, respectively. Sex distribution (*p* = 0.526), age (*p* = 0.232), family history (*p* = 0.552), preceding URI (*p* = 0.448), PASI (*p* = 0.537), presence of pruritus (*p* = 0.420), duration of disease (*p* = 0.101), epidermal thickness (*p* = 0.559), horny layer thickness (*p* = 0.056), rete ridge count (*p* = 0.351), and grade of papillary dermal vessel dilatation (*p* = 0.129) were not significantly different between the two groups. The grade of inflammatory cellular infiltration in the dermal PD-1-high group (mean ± SD, 2.15 ± 0.55) was greater than that in the dermal PD-1-low group (mean ± SD, 1.63 ± 0.72) (*p* = 0.042).

### 3.2. mRNA Expression Levels of S100A8 and PD-L1 of CPP Lesions Compared to Non-Lesions and PD-L1 Immunohistochemisry in GP

As IHC staining for PD-L1 could not be clearly interpreted, the mRNA expression levels of PD-L1 in the lesional and non-lesional skin samples of 11 patients with CPP were comparatively analyzed. Compared with those in the non-lesional skin, the mRNA expression levels of S100A8 were upregulated in the lesional skin, indicating the lesional versus non lesional skin sampling was properly performed. (Figure 1H). In fact, the mRNA level of PD-L1 in the lesional skin was significantly higher than that in non-lesional skin (*p* = 0.002) (Figure 1H).

In guttate psoriasis, the IHC for PD-L1 was performed in the lesional skin. The dermal PD-1-low group less frequently showed PD-L1 positivity, while the dermal PD-1-high group more frequently showed PD-L1 positivity. All patients of the dermal PD-1-high group expressed PD-L1 in the epidermis. 

### 3.3. Clinicoprognostic and Histopathological Characteristics of GP according to the PD-1 Expression Levels

The mean age of 33 patients (19 men and 14 women) with GP was 24.61 years (range, 5–56 years). Of these 33 patients, 14 (42.4%) and 19 (57.6%) were assigned to the dermal PD-1-low (Figure 2A) and dermal PD-1-high (Figure 2B) groups, respectively. The T cell landscape in the representative patients with GP in dermal PD-1-high group is illustrated in Appendix A. While the majority of T cells in the epidermis were positive for CD8, the majority of T cells in the dermis were positive for CD4, expressing PD-1 together.

The demographic data and clinical characteristics of patients with GP are summarized in Table 3. Sex distribution was not significantly different between the groups (*p* = 0.966). The age of patients in the dermal PD-1-low group was not significantly lower than that of patients in the dermal PD-1-high group (*p* = 0.073). A family history of psoriasis was not significantly different between the two groups (*p* = 0.424); only one patient had a family history of psoriasis. The number of patients with URI symptoms before the onset of GP was not significantly different between the dermal PD-1-low (8 patients (57.1%)) and dermal PD-1-high groups (14 patients (73.7%)) (*p* = 0.388). The PASI scores in the dermal PD-1-low group (mean ± SD, 7.46 ± 4.83) was not significantly higher than those in the dermal PD-1-high group (mean ± SD, 6.02 ± 3.88) (*p* = 0.388). The BSA in the dermal PD-1-low group (mean ± SD, 9.96 ± 6.33%) was not significantly higher than that of the dermal PD-1-low group (mean ± SD, 7.13 ± 6.23%) (*p* = 0.254). The prevalence of pruritus was not significantly different between the groups (*p* = 0.561).

Disease duration in the dermal PD-1-low group (mean ± SD, 18.57 ± 25.24 months) was significantly higher than that in the dermal PD-1-high group (mean ± SD, 6.53 ± 8.59 months) (*p* = 0.002) (Figure 2C). Additionally, disease duration was longer than 4 months in 12 (85.7%) and 8 (42.1%) patients belonging to the dermal PD-1-low and dermal PD-1-high groups, respectively (*p* = 0.011). Time to disease resolution in the dermal PD-1-low group (mean ± SD, 17.36 ± 25.83 months) was significantly higher than that in the dermal PD-1-high group (mean ± SD, 5.32 ± 8.53 months) (*p* = 0.008) (Figure 2D).

The histopathological characteristics of patients with GP are summarized in Table 4. The epidermal thickness in the dermal PD-1-low group (mean ± SD, 210.92 ± 43.02 µm) was significantly greater than that in the dermal PD-1-high group (mean ± SD, 171.96 ± 48.13 µm) (*p* = 0.046) (Figure 2E). The horny layer thickness (*p* = 0.199), rete ridge count (*p* = 0.900), grade of inflammatory cellular infiltration (*p* = 0.651), and grade of papillary dermal vessel dilatation (*p* = 0.892) were not significantly different between the two groups.

To determine RFS, the follow-up period ranged from 2 to 228 months (median, 69 months). During the follow-up period, the incidence of overall psoriasis in the dermal PD-1-low group was not significantly higher than in the dermal PD-1-high group (p = 0.062). The GP relapse rate was not significantly different between the groups (*p* = 0.803). However, the incidence of plaque psoriasis relapse in the dermal PD-1-low group (5 patients (35.7%)) was significantly higher than that in the dermal PD-1-high group (0 patients (0.0%)) (*p* = 0.005) (Table 3). The median RFS period of the study cohort was 72 months (95% confidence interval (CI), 51.4–92.6 months). The RFS (*p* = 0.279) (Figure 2F) and RFS-G were not significantly different between the groups (*p* = 0.844) (Figure 2G). However, the RFS-P in the dermal PD-1-high group was significantly higher than that in the dermal PD-1-low group (*p* = 0.016) (Figure 2H).

## 4. Discussion

CD4+Foxp3+ Treg cells are involved in the pathogenesis of psoriasis [12]. Additionally, several immune checkpoint proteins regulate the pathogenesis of psoriasis by modulating the inflammatory immune responses mediated by dendritic, Th17, and Th1 cells. The level of CTLA4, a traditional immune checkpoint molecule that inhibits T cell CD28(B7-1) costimulatory function, was reported to be negatively correlated with the severity of psoriasis. In mild psoriatic lesions, the membrane level of CTLA4 is upregulated. The inhibition of CTLA4 function exacerbates psoriatic lesions [13]. Recently, a deficiency in V-domain immunoglobulin suppressor of T cell activation (VISTA), an immune checkpoint molecule that belongs to the B7 family, was reported to aggravate psoriasiform dermatitis in a murine model of imiquimod-induced psoriasis [14].

The inhibition of PD-1 and PD-L1, which are the most widely investigated immune checkpoint molecules, is critical for the efficacy of cancer treatment. PD-1 plays a critical role in the pathogeneses of various malignancies, including melanoma [15], especially in tumor immune escape [16]. Psoriasiform skin eruption is one of the cutaneous side effects of PD-1 inhibitors [17]. Accordingly, anti-PD-1/PD-L1-induced psoriasis is a topic of interest in the field of dermato-oncology.

Various studies have examined the correlation between PD-1 expression and CPP [3,4,5,6,7,18]. PD-1, a cell surface membrane receptor expressed on various inflammatory cells, transduces inhibitory signals to immune cells, including effector T cells, and promotes Treg activity [3,19]. Previous studies have reported the correlation between PD-1 and various autoimmune diseases, including rheumatoid arthritis, type I diabetes, multiple sclerosis [20], and psoriasis. PD-1 signaling is also involved in the pathogenesis of cutaneous diseases, such as allergic contact dermatitis and cutaneous graft-versus-host-disease [21,22].

Only limited studies have examined the correlation between PD-1 expression and the clinicoprognostic and distinct clinicopathological characteristics of CPP and GP.

In this study, the upregulated expression of PD-1 was associated with the severity of CPP. Further, the epidermis in the epidermal PD-1-high group was found to be thicker than that in the epidermal PD-1-low group. The epidermal PD-1-high group also had more prominent vessel dilatation than the epidermal PD-1-low group. Compared with those in the epidermal PD-1-low group, PASI score and disease duration were significantly higher in the epidermal PD-1-high group. Such a finding can be attributed to the compensatory upregulation of PD-1 to overcome the Th17 and Th22 pathways, which are critical for epidermal proliferation in CPP [4,23].

Although PD-1 is critical for immune exhaustion (which regulates immune tolerance), PD-1-positive T cells exhibit effector function rather than exhaustion in cases of chronic inflammation [24]. The expression of PD-1 is upregulated in antigen-specific T cells, which exhibit effector functions in the lesions of CPP (a chronic inflammatory condition). Thus, the upregulated epidermal PD-1 expression in CPP may suggest active and severe lesions, which are associated with enhanced PASI scores, increased disease duration, and thick epidermis.

Contradictory findings have been reported regarding the correlation between PD-1 expression and the severity of CPP and GP. In CPP, epidermal T cells mostly consist of CD8+ T cells, whereas dermal T cells mostly consist of CD4+ T cells. Yan et al. [25]. reported the distinct regulatory mechanism of IL17 and IFN-γ in CPP and GP. The cytokines derived from CD8+ T cells play a critical role in CPP. Epidermal CD8+IL17+ cells are involved in the immunopathogenesis of CPP. However, the role of epidermal CD8+IL17+ cells in the pathogenesis of GP is unclear. In this study, PD-1-positive cells were not detected in the epidermis of patients with GP, except in one case. Therefore, the expression of PD-1 in epidermal T cells may determine the clinicoprognostic characteristics of CPP.

Some studies have comparatively analyzed the distinct immunopathogeneses of GP and CPP. The initiation of GP was associated with cross-reactivity between streptococcal M protein and epidermal autoantigens, such as keratin 17 presented by HLA-Cw6 molecules [26]. The levels of CD4+IL17+ cells in patients with GP were higher than those in patients with CPP. This finding can be attributed to the decreased levels of Treg cells, which cannot inhibit the IL6 and CD4+IL17 immune axis responses in GP [25,27]. In GP, the downregulated dermal PD-1 expression was associated with poor clinical outcomes, prolonged clinical course, and enhanced epidermal thickness. The proportion of patients with disease duration longer than 4 months in the dermal PD-1-low group (85.7%) was significantly higher than that in the dermal PD-1-high group (42.1%) (*p* = 0.011). Compared with those in the PD-1-high group, time to disease resolution (a marker of treatment response) and the rate of disease progression to plaque psoriasis were higher in the dermal PD-1-low group.

The role of disrupted immune tolerance in the pathogenesis of GP (acute or subacute unstable skin lesions) may be more important than that in the pathogenesis of CPP (chronic, life-long, stable skin lesions). GP is a type of acute rapidly-involuting inflammatory disease associated with a specific antigenic insult. In GP, PD-1 expression may lead to T cell exhaustion and may consequently decrease the severity of inflammation. Therefore, downregulated dermal PD-1 expression in GP indicates the impaired suppression of acute inflammation, which leads to severe histopathological changes and poor prognosis, thereby requiring routine follow-up for the early detection of disease progression to CPP.

Several pathological mechanisms have been proposed for the role of PD-1 in the pathogenesis of psoriasis. Joanna et al. [3]. reported that the mRNA levels of *PD-1* in patients with psoriasis were significantly lower than those in healthy controls. Further, another study reported that the peripheral blood levels of PD-1-positive CD4+ T cells and CD8+ T cells in the psoriasis group were markedly lower than those in the healthy control group [18]. Ryota et al. [6] demonstrated that PD-1 deficiency exacerbates psoriatic inflammation and activated CD8+ T cells in the epidermis produce IL6. The downregulation of PD-1 results in the upregulation of INFG, TNF-α, and IL17. Such a finding suggests that PD-1 regulates the Th1 and Th17 signaling pathways, which mediate the pathogenesis of psoriasis [4,5]. A recent study reported that the upregulated levels of PD-1 in CD8+CD103 T cells in the psoriatic epidermis were correlated with disease severity and histopathological changes [28]. Additionally, treatment with the PD-1 crystallizable fragment alleviated psoriatic inflammation and exerted additive therapeutic effects with anti-TNF-α therapy [7].

The results of this study and previous studies indicate that the membrane expression of PD-1 in T cells modulates the immune response and the production of cytokines, which are critical for the pathogenesis of psoriasis. To our knowledge, this is the first study to demonstrate that the function of PD-1 may vary depending on the clinical type of psoriasis. However, further studies with a large cohort are needed to verify this finding.

As IHC staining for PD-L1 could not be clearly interpreted, we comparatively analyzed the mRNA expression of PD-L1 of the epidermal tissues obtained from 11 patients with CPP. In this study, the mRNA levels of *PD-L1* in the lesional skin were significantly higher than those in the non-lesional skin of patients with CPP. The membrane expression of PD-L1 in keratinocytes or dendritic cells and macrophages may be upregulated to suppress the enhanced immune response [29]. This additional finding may indicate enhanced immune response in PD-1-positive T cells in the context of chronic inflammation, such as CPP. However, some previous studies have reported that the expression of PD-L1 is downregulated in psoriatic epidermis [30]. As previous studies did not focus on the clinical features of psoriasis, such as duration of disease and types of psoriasis, a gap may exist between the results of this study and those of previous studies. Further studies with a large sample size that also perform quantitative analysis of PD-L1 mRNA according to clinicoprognostic and histopathologic features of CPP and GP are needed.

This study had several limitations. First, this study is of a retrospective nature and was performed at a single medical center in South Korea with a relatively small sample size. Second, the expression of PD-1 (IHC) was determined using semi-quantitative scoring methods. Finally, a quantitative analysis of *PD-L1* mRNA was not performed in this study as the size of the tissue sample was less than 1 mm. Therefore, prospective studies with a large sample size and long follow-up duration involving subjects from both Asian and Western populations must be performed to determine the correlation between PD-1 expression and the clinicoprognostic characteristics of CPP and GP.

## 5. Conclusions

In conclusion, the upregulated expression of epidermal PD-1 was correlated with the chronicity and severity of CPP while the downregulated expression of dermal PD-1 was correlated with poor prognosis of GP.

## Figures and Tables

**Figure 1 jcm-10-05200-f001:**
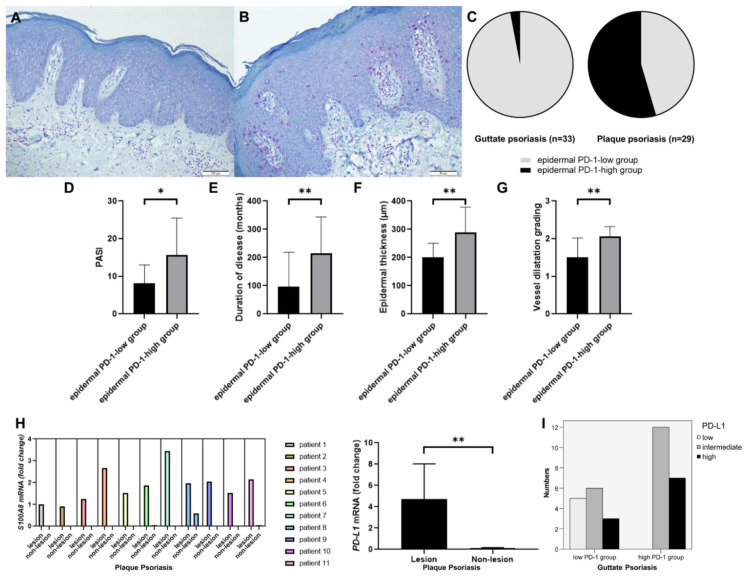
Clinical and histopathological characteristics of patients with chronic plaque psoriasis (CPP) in the PD-1-high and PD-1-low groups. (**A**–**C**) Of the 29 patients, 14 and 15 patients were assigned to the epidermal PD-1-low and epidermal PD-1-high groups, respectively. Among the 33 patients with GP, epidermal PD-1 expression was only detected in 1 patient. (**D**) The PASI scores in the epidermal PD-1-high group was significantly higher than that in the epidermal PD-1-low group (*p* = 0.014). (**E**) Disease duration in the epidermal PD-1-high group was significantly higher than that in the epidermal PD-1-low group (*p* = 0.009). (**F**) The thickness of the epidermis in the epidermal PD-1-high group was significantly greater than that in the epidermal PD-1-low group (*p* = 0.004). (**G**) The grade of vessel dilatation in the epidermal PD-1- high group was significantly higher than that in the epidermal PD-1-low group (*p* = 0.002). (**H**) Compared with those in the non-lesional skin, the mRNA expression levels of S100A8 were upregulated in the lesional skin. The mRNA level of PD-L1 in lesional skin was significantly higher than that in non-lesional skin (*p* = 0.002). (**I**) The level of IHC positivity of PD-L1 in the dermal PD-1-low group of guttate psoriasis was significantly lower than that of dermal PD-1-high group (*p* = 0.03, Fisher’s exact test). * *p* < 0.05, ** *p* < 0.01.

**Figure 2 jcm-10-05200-f002:**
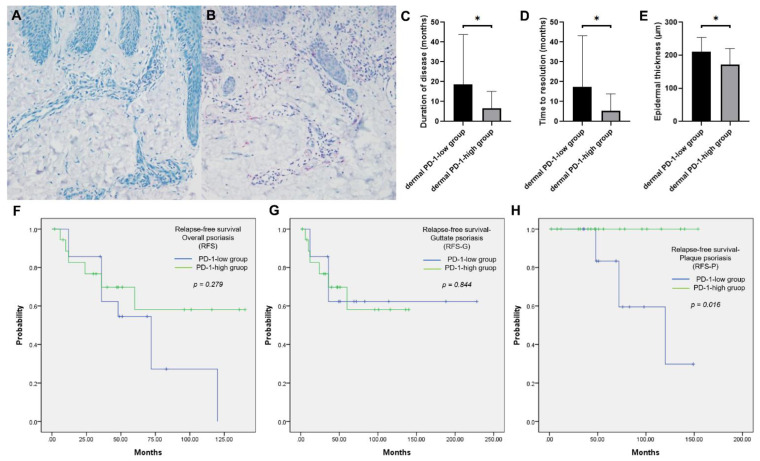
Clinicoprognostic and histopathological characteristics of patients with guttate psoriasis in the dermal PD-1-high and dermal PD-1-low groups. (**A**,**B**) Of the 33 patients, 14 and 19 were assigned to the dermal PD-1-low and dermal PD-1-high groups, respectively. (**C**) Disease duration in the dermal PD-1-low group was significantly higher than that in the dermal PD-1-high group (*p* = 0.002). **(D)** Time to disease resolution in the dermal PD-1-low group was significantly higher than that in the dermal PD-1-high group (*p* = 0.008). (**E**) Epidermal thickness in the dermal PD-1-low group was significantly greater than that in the dermal PD-1-high group (*p* = 0.046). (**F**) Relapse-free survival (RFS) was not significantly different between the groups (*p* = 0.279). (**G**) RFS of patients with guttate psoriasis (RFS-G) was not significantly different between the groups (*p* = 0.844); (**H**) RFS of patients with plaque-type psoriasis (RFS-P) in the dermal PD-1-high group was significantly higher than that in the dermal PD-1-low group (*p* = 0.016). * *p* < 0.05.

**Table 1 jcm-10-05200-t001:** Clinical characteristics of patients with chronic plaque psoriasis according to the epidermal expression levels of PD-1.

Characteristics	Epidermal PD-1-Low Group (*n* = 14)	Epidermal PD-1-High Group (*n* = 15)	*p-*Value
Sex (n (%))			0.298
Male	9 (64.3)	12 (80.0)	
Female	5 (42.9)	3 (20.0)	
Age (years)			0.949
Range	12–81	17–66	
Mean ± SD	45.64 ± 19.17	46.33 ± 15.15	
Family history of psoriasis			0.483
Yes	1 (7.1)	0 (0.0)	
No	13 (92.9)	15 (100.0)	
Preceding upper respiratory infection			0.483
Yes	1 (7.1)	0 (0.0)	
No	13 (92.9)	15 (100.0)	
PASI score			0.014 *
Range	1.2–19.8	1.6–41.8	
Mean ± SD	8.20 ± 4.83	15.71 ± 9.77	
Pruritus			0.125
Yes	14 (100.0)	12 (80.0)	
No	0 (0.0)	3 (20.0)	
Disease duration (months)			0.009 *
Range	1–360	36–480	
Mean ± SD	96.79 ± 122.0	215.2 ± 128.2	

* Statistically significant. Abbreviations: PD-1, programmed cell death protein-1; SD, standard deviation; PASI, psoriasis area and severity index.

**Table 2 jcm-10-05200-t002:** Histopathological characteristics of patients with chronic plaque psoriasis according to the epidermal expression level of PD-1.

Characteristics	Epidermal PD-1-Low Group (*n* = 14)	Epidermal PD-1-High Group (*n* = 15)	*p-*Value
Epidermal thickness (µm)			0.004 *
Range	127.92–293.11	157.38–475.04	
Mean ± SD	200.19 ± 20.74	289.88 ± 88.88	
Horny layer thickness (µm)			0.201
Range	11.63–117.40	26.83–163.65	
Mean ± SD	44.23 ± 29.11	58.59 ± 36.33	
Rete ridge count (n)			0.354
Range	8–15	8–15	
Mean ± SD	11.43 ± 2.06	12.20 ± 1.70	
Cellular infiltration grading			0.567
Range	1–3	1–3	
Mean ± SD	1.79 ± 0.80	1.93 ± 0.59	
Vessel dilatation grading			0.002 *
Range	1–2	2–3	
Mean ± SD	1.50 ± 0.52	2.07 ± 0.26	

* Statistically significant. Abbreviations: PD-1, programmed cell death protein-1; SD, standard deviation.

**Table 3 jcm-10-05200-t003:** Clinical characteristics of patients with guttate psoriasis according to the dermal expression level of PD-1.

Characteristics	Dermal PD-1-Low Group (*n* = 14)	Dermal PD-1-High Group (*n* = 19)	*p-*Value
Sex (n (%))			0.966
Male	8 (57.1)	11 (57.9)	
Female	6 (42.9)	6 (42.9)	
Age (years)			0.086
Range	5–47	9–56	
Mean ± SD	20.86 ± 12.56	27.37 ± 11.61	
≤18	8 (57.1)	5 (26.3)	0.073
>18	6 (42.9)	14 (73.7)	
Family history of psoriasis			0.424
Yes	1 (7.1)	0 (0.0)	
No	13 (92.9)	19 (100.0)	
Preceding upper respiratory infection			0.319
Yes	8 (57.1)	14 (73.7)	
No	6 (42.9)	5 (26.3)	
PASI score			0.388
Range	0.90–15.30	2.00–14.50	
Mean ± SD	7.46 ± 4.83	6.02 ± 3.88	
BSA (%)			0.254
Range	1.00–20.00	1.00–19.00	
Mean ± SD	9.96 ± 6.33	7.13 ± 6.23	
Pruritus			0.561
Yes	9 (64.3)	14 (73.7)	
No	5 (35.7)	5 (26.3)	
Disease duration (months)			0.002 *
Range	3–83	1–40	
Mean ± SD	18.57 ± 25.24	6.53 ± 8.59	
≥4	2 (14.3)	11 (57.9)	0.011*
>4	12 (85.7)	8 (42.1)	
Time to disease resolution (months)			0.008 *
Range	1–83	1–40	
Mean ± SD	17.36 ± 25.83	5.32 ± 8.53	
Relapse of overall psoriasis			0.062
Yes	9 (64.3)	6 (31.6)	
No	5 (35.7)	13 (68.4)	
Relapse of guttate psoriasis			0.803
Yes	5 (35.7)	6 (31.6)	
No	9 (64.3)	13 (68.4)	
Relapse of plaque psoriasis			0.005 *
Yes	5 (35.7)	0 (0.0)	
No	9 (64.3)	19 (100.0)	

* Statistically significant. Abbreviations: PD-1, programmed cell death protein-1; SD, standard deviation; PASI, psoriasis area and severity index; BSA, body surface area.

**Table 4 jcm-10-05200-t004:** Histopathological characteristics of patients with guttate psoriasis according to the dermal expression level of PD-1.

Characteristics	Dermal PD-1-Low Group (*n* = 14)	Dermal PD-1-High Group (*n* = 19)	*p-*Value
Epidermal thickness (µm)			0.046 *
Range	154.28–295.05	92.63–261.91	
Mean ± SD	210.92 ± 43.02	171.96 ± 48.13	
Horny layer thickness (µm)			0.199
Range	23.27–97.70	31.50–112.55	
Mean ± SD	55.76 ± 18.31	50.54 ± 19.23	
Rete ridge count (n)			0.900
Range	6–14	8–14	
Mean ± SD	11.07 ± 2.40	11.11 ± 2.00	
Cellular infiltration grading			651
Range	1–2	1–3	
Mean ± SD	1.50 ± 0.52	1.63 ± 0.60	0.
Vessel dilatation grading			0.892
Range	1–3	1–3	
Mean ± SD	1.86 ± 0.66	1.74 ± 0.65	

* Statistically significant. Abbreviations: PD-1, programmed cell death protein-1; SD, standard deviation.

## Data Availability

The data that support the findings of this study are available from the corresponding author upon reasonable request.

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
