# Peer review of "Clinicoprognostic and Histopathological Features of Guttate and Plaque Psoriasis Based on PD-1 Expression"

_jcm, 2021, doi:10.3390/jcm10215200_

Round 1

Reviewer 1 Report

Jung et al. studied the clinicopathological features between guttate psoriasis and chronic plaque-type psoriasis focusing on PD-1 expression of lesional infiltrating inflammatory cells. The variety of PD-1 expression level in different psoriasis subtypes possibly explains distinct pathomechanism between the subtypes.

  1. To speculate a possible role of PD-1 expression in inflammatory cells, PD-L1 expression level can be one of the most essential issues. Please provide the data presenting PD-L1 expression discussed in “discussion part”.
  2. While the ratio of PD-1 expressing cells is presented, is there any difference of inflammatory cell counts between GP and CPP?
  3. As a result, only the ratio of PD-1 expression in inflammatory cells is presented in this study. What type of cells mainly expresses PD-1 in lesional psoriatic skin? As recently shown, CD8-positive resident cells can be a source of the PD-1 expression. If so, stronger PD-1 expression in longer and more persistent psoriasis lesion is reasonable.
  4. Minor points. Please remove unnecessary line feeds in the reference list.

Author Response

Dear Editor

Journal of Clinical Medicine

We would like to submit our revised manuscript entitled “Clinicoprognostic and histopathological features of guttate and plaque psoriasis based on PD-1 expression”. We would like to thank the reviewers for their critical comments and constructive suggestions. The manuscript has been rewritten and substantially improved in quality in response to reviewers’ concerns. All changes made to the manuscript are shown in red font.

Below please find our detailed responses to the reviewers’ comments, including modifications and revisions to the manuscript.

Thank you for your ime and consideration.

Sincerely,

Sung Eun Chang, MD PhD

Department of Dermatology, Asan Medical Center, University of Ulsan College of Medicine, 88 Olympic-ro 43 gil, Songpa-gu, Seoul 05505, Korea

Tel: +82-2-3010-3460; Fax: +82-2-486-7831; E-mail: [email protected]

Reviewer(s)’ Comments to Author

  • Reviewer 1
  1. To speculate a possible role of PD-1 expression in inflammatory cells, PD-L1 expression level can be one of the most essential issues. Please provide the data presenting PD-L1 expression discussed in “discussion part”.

=> Thank you for your considerate comment. We’re so sorry but as we mentioned in the section of the result 3.2, IHC staining for PD-L1 could not be clearly interpreted in that we could not find any differences between IHC staining for PD-L1 of keratinocyte with naked eye. Instead, the mRNA expression levels of PD-L1 in the lesional and non-lesional skin samples of 11 patients with CPP were comparatively analyzed. In the discussion section, we suggested that upregulated mRNA levels of PD-L1 in lesional skin compared with non-lesional skin of patients with CPP may advocate enhanced immune response in PD-1-positive T cells in the context of chronic inflammation, such as CPP.

  1. While the ratio of PD-1 expressing cells is presented, is there any difference of inflammatory cell counts between GP and CPP?

=> Thank your for your sharp insight. There was no difference between the grade of inflammatory cell infiltration between GP and CPP (mean ± SD, 1.67 ± 0.55 vs. 1.86 ± 0.68, p > 0.05).

  1. As a result, only the ratio of PD-1 expression in inflammatory cells is presented in this study. What type of cells mainly expresses PD-1 in lesional psoriatic skin? As recently shown, CD8-positive resident cells can be a source of the PD-1 expression. If so, stronger PD-1 expression in longer and more persistent psoriasis lesion is reasonable.

=> Thank your for your keen insight. We added IHC analysis for representative cases of GP and CPP using anti-CD4 and CD8 antibodies to evaluate immunologic milieu specifying PD-1-expressing T cells in psoriatic lesions. As you pointed out, in the representative patients with CPP in epidermal PD-1-high group, we found that while the majority of the T cells in dermis were positive for CD4, the majority of T cells in epidermis were positive for CD8, expressing PD-1 together. We also added supplementary figure 1 and 2.

  1. Minor points. Please remove unnecessary line feeds in the reference list.

=>Thank you for your kind pointing out. We revised the form of the reference according to your journal, Journal of Clinical Medicine.

Reviewer 2 Report

The manuscript by Jung et al. provides an interesting assessment of histopathological differences between high and low PD-1 expressing patients with chronic plaque psoriasis and guttate psoriasis. Overall, the authors find that chronic plaque patients with high epidermal PD-1 expression had more severe features, while almost all patients with guttate psoriasis had low epidermal PD-1 expression. Guttate psoriasis patients with high PD-1 expression in dermis had overall less severe disease than those with low PD-1 expression. Lastly, a retrospective analysis of relapse-free survival is shown and the authors observe higher relapse incidence in patients with low dermal PD-1 as compared to those with high dermal PD-1 expression. 

The work is well conducted, addressing an interesting question, and the manuscript is well written. Despite the need for this work to be published, this reviewer believes that one important aspect needs to be addressed first. The manuscript claims to assess clinicoprognostic and histopathological features of psoriasis based on PD-1 expression. The authors should provide evidence that the observations and new advancements provided are not simply a reflection on the (well-described) biology of T-cells in psoriasis. If instead that is indeed the case, then this should be reflected in the in the manuscript. Please see the comments below. 

Major comments: 

  • The authors show that the patient group with higher epidermal PD-1 staining have more severe histopathological features. It is known from many sources of evidence that epidermal T-cells are correlated with disease severity, and that upon successful treatment epidermal T-cell numbers decrease early on. While PD-1 is not exclusively expressed in T-cells, they are likely among the main cell types that express it in epidermis. Could the authors verify whether the association of high epidermal PD-1 expression is linked to increased T-cell presence in more severe disease? In other words, is the PD-1 difference due to increased T-cell numbers in the epidermis or is it due to an increased expression in similar overall numbers of T-cells? 

  • In complementing the question above, how is the T-cell landscape in those patients with guttate psoriasis which have mostly low epidermal PD-1 expression? In other words, could the authors show T-cell presence, and ideally in a double staining with PD-1. 

  • The authors show PDL1 mRNA expression as a surrogate for PD-L1 staining in 11 patients (due to difficulties with the staining of the protein), including both lesional and non-lesional skin (Figure 1I). While this comparison underscores the inherent differences in cellularity between healthy and psoriasis skin, it does not add further information on the nature of the PD-1 / PD-L1 interaction. How does PDL1 expression compare between high PD-1 patients and low PD-1 patients? 

Minor comments: 

  • The authors show histopathologic differences between patients with low and those that have high epidermal PD-1. Among these they describe a marginal non-significant increase in the stratum corneum in Table 2. In Figure 1A and B, the authors display representative histology sections of a patient with low epidermal PD-1 (A) and one with high epidermal PD-1 (B). From panel B there is clearly discernible parakeratosis, whereas in panel A this is not easily discernible. Do the authors see differences in metrics for parakeratosis between the two groups? 
  • In line 312 the authors state that only a limited number of studies have addressed the question tackled by the present manuscript. Could the authors please find a way to reference these works? 
  • In the discussion, the paragraph starting on line 339 addresses an important distinction between plaque psoriasis and guttate psoriasis: apparently opposite outcomes from high epidermal PD-1 and high dermal PD-1. Instead of re-iterating the findings, could the authors further elaborate on the underlying differences between guttate and plaque psoriasis that could consolidate this seemingly opposite finding, where published findings are available? 

Author Response

Dear Editor

Journal of Clinical Medicine

We would like to submit our revised manuscript entitled “Clinicoprognostic and histopathological features of guttate and plaque psoriasis based on PD-1 expression”. We would like to thank the reviewers for their critical comments and constructive suggestions. The manuscript has been rewritten and substantially improved in quality in response to reviewers’ concerns. All changes made to the manuscript are shown in red font.

Below please find our detailed responses to the reviewers’ comments, including modifications and revisions to the manuscript.

Thank you for your ime and consideration.

Sincerely,

Sung Eun Chang, MD PhD

Department of Dermatology, Asan Medical Center, University of Ulsan College of Medicine, 88 Olympic-ro 43 gil, Songpa-gu, Seoul 05505, Korea

Tel: +82-2-3010-3460; Fax: +82-2-486-7831; E-mail: [email protected]

Reviewer(s)’ Comments to Author

  • Reviewer 2
  1. The authors show that the patient group with higher epidermal PD-1 staining have more severe histopathological features. It is known from many sources of evidence that epidermal T-cells are correlated with disease severity, and that upon successful treatment epidermal T-cell numbers decrease early on. While PD-1 is not exclusively expressed in T-cells, they are likely among the main cell types that express it in epidermis. Could the authors verify whether the association of high epidermal PD-1 expression is linked to increased T-cell presence in more severe disease? In other words, is the PD-1 difference due to increased T-cell numbers in the epidermis or is it due to an increased expression in similar overall numbers of T-cells? 

è Thank you for your keen insight. We added IHC analysis for representative cases of GP and CPP using anti-CD4 and CD8 antibodies to evaluate immunologic milieu specifying PD-1-expressing T cells in psoriatic lesions. In the representative patients with CPP in epidermal PD-1-high group, we found that while the majority of the T cells in dermis were positive for CD4, the majority of T cells in epidermis were positive for CD8, expressing PD-1 together. We added this result in the section of 3.1 and illustrated supplementary figure 1.

  1. In complementing the question above, how is the T-cell landscape in those patients with guttate psoriasis which have mostly low epidermal PD-1 expression? In other words, could the authors show T-cell presence, and ideally in a double staining with PD-1. 

è Thank you for your keen insight. We added IHC analysis for representative cases of GP and CPP using anti-CD4 and CD8 antibodies to evaluate immunologic milieu specifying PD-1-expressing T cells in psoriatic lesions. In the representative patients with GP in dermal PD-1-high group, we found that while the majority of the T cells in epidermis were positive for CD8, the majority of T cells in dermis were positive for CD4, expressing PD-1 together. We added this result in the section of 3.3 and illustrated supplementary figure 2.

  1. The authors show PDL1mRNA expression as a surrogate for PD-L1 staining in 11 patients (due to difficulties with the staining of the protein), including both lesional and non-lesional skin (Figure 1I). While this comparison underscores the inherent differences in cellularity between healthy and psoriasis skin, it does not add further information on the nature of the PD-1 / PD-L1 interaction. How does PDL1expression compare between high PD-1 patients and low PD-1 patients? 

è Thank you for keen point. PD-1 signaling negatively regulates T cell-mediated immune responses by binding to PD-L1. If the expression of PD-L1 is too low in the lesion of CPP, there might be no clincal and histopathological differences between epidermal PD-1-low group and epidermal PD-1-high group. Therefore, we comparatively analyzed the mRNA expression of PD-L1 of the epidermal tissues obtained from 11 patients with CPP, to find the mRNA levels of PD-L1 in lesional skin were significantly higher than those in the non-lesional skin of patients with CPP. Through this, the membrane expression of PD-L1 in keratinocytes or dendritic cells and macrophages may be upregulated to suppress the enhanced immune response, advocating enhanced immune response in PD-1-positive T cells in the context of chronic inflammation.

  1. The authors show histopathologic differences between patients with low and those that have high epidermal PD-1. Among these they describe a marginal non-significant increase in the stratum corneum in Table 2. In Figure 1A and B, the authors display representative histology sections of a patient with low epidermal PD-1 (A) and one with high epidermal PD-1 (B). From panel B there is clearly discernible parakeratosis, whereas in panel A this is not easily discernible. Do the authors see differences in metrics for parakeratosis between the two groups? 

è Thank you for sharp pointing out. As we mentioned in the section of method, we evaluated only 5 histopathologic charatcteristics including epidermal thickness, horny layer thickness, rete ridge count, grade of inflammatory cellular infiltration, and grade of papillary dermal vessel dilatation which were reported to be related to the clinical severity of psoriasis by Kim et al. Therefore, we didn’t evaluate parakeratosis between the two groups.

  1. In line 312 the authors state that only a limited number of studies have addressed the question tackled by the present manuscript. Could the authors please find a way to reference these works? 

è Thank you for your kind advice. Unfortunately, we could not find line number in the manuscript. If you let us know the sentence or paragraph you’re mentioning, we’ll revise as soon as possible. We’re so sorry.

  1. In the discussion, the paragraph starting on line 339 addresses an important distinction between plaque psoriasis and guttate psoriasis: apparently opposite outcomes from high epidermal PD-1 and high dermal PD-1. Instead of re-iterating the findings, could the authors further elaborate on the underlying differences between guttate and plaque psoriasis that could consolidate this seemingly opposite finding, where published findings are available? 

è Thank you for your kind advice. Unfortunately, we could not find line number in the manuscript. If you let us know the sentence or paragraph you’re mentioning, we’ll revise as soon as possible. We’re so sorry.

Round 2

Reviewer 1 Report

I appreciate the revision.

I also understand the difficulty on PD-L1 IHC. However, I think the data of PD-L1 expression is indispensable in guttate psoriasis. Please provide the data of PD-L1 mRNA expression in the GP lesion comparing with non-lesion.